# GloUCP: A global 1 km spatially continuous urban canopy parameters for the WRF model

Weilin Liao[1,2], Yanman Li[1], Xiaoping Liu[1,3], Yuhao Wang[1], Yangzi Che[1], Ledi Shao[1], Guangzhao Chen[1], Hua Yuan[3,4], Ning Zhang[5], Fei Chen[6]

[1]Guangdong Key Laboratory for Urbanization and Geo-simulation, School of Geography and Planning, Sun Yat-sen University, Guangzhou, 510275, China

[2]Key Laboratory of Urban Meteorology, China Meteorological Administration, Beijing, 100089, China

[3]Southern Marine Science and Engineering Guangdong Laboratory (Zhuhai), Zhuhai, 519082, China

[4]School of Atmospheric Sciences, Sun Yat-sen University, Guangzhou, 510275, China

[5]School of Atmospheric Sciences, Nanjing University, Nanjing, 210023, China

[6]Division of Environment and Sustainability, The Hong Kong University of Science and Technology, Hong Kong, 999077, China

*Correspondence to*: Xiaoping Liu (liuxp3@mail.sysu.edu.cn)

**Abstract.** The complexities of urban climate and environmental challenges have garnered significant attention in the 21st century. Numerical simulations, offering high spatiotemporal resolution meteorological data, are essential tools in meteorological research and atmospheric science. Accurate representation of urban morphology parameters is crucial for enhancing the precision of these simulations in urban areas. Despite the availability of urban canopy parameter (UCP) data
for 44 major cities in the United States and 60 in China for the weather research and forecasting (WRF) model, a comprehensive global dataset representing urban morphology remains absent. This study addresses this gap by leveraging existing global three-dimensional vector data of buildings, including footprints and heights, to compile a global 1 km spatially continuous UCP (GloUCP) dataset for the WRF model. Our findings indicate that GloUCP not only surpasses existing datasets in accuracy but also provides superior spatial coverage. In key urban agglomerations such as Beijing-
Tianjin-Hebei, the Yangtze River Delta, and the Guangdong-Hong Kong-Macao Greater Bay Area in China, GloUCP offers detailed and reliable urban morphological information that closely aligns with reference datasets, outperforming other available sources. Similarly, in U.S. cities like Seattle, San Francisco, and Philadelphia, GloUCP consistently achieves lower RMSE values and higher correlation coefficients, demonstrating its robustness in modeling diverse urban environments. Furthermore, GloUCP's capability to effectively capture the vertical distribution of buildings, particularly in high-rise areas,

highlights its utility in urban climate modeling and related applications. As UCPs are pivotal in regulating atmospheric responses to urbanization, the availability of this globally consistent urban description is a crucial prerequisite for advancing model development and informing climate-sensitive urban planning policies. The GloUCP dataset, converted to WRF binary file format, is available for download at https://doi.org/10.6084/m9.figshare.27011491 (Liao et al., 2024).

## 1 Introduction

Cities play a crucial role in driving climate change, serve as hotspots for climate impacts, and are central to climate solutions (Zhao et al., 2021; Liu et al., 2020). The complexity of urban environments, combined with the limited availability of urban-specific observations, makes it imperative to rely on models to simulate urban processes and their interactions with regional and global climates (Oleson et al., 2011). Therefore, numerical models are indispensable for understanding future urban climate scenarios and for informing policy and planning (Chen et al., 2011).

In recent years, regional climate modeling has increasingly focused on hyper-resolution simulations, which aim to resolve land surface processes at scales of 1 km or finer (Li et al., 2024). This shift towards finer resolutions is driven by the need for more accurate operational forecasts, particularly in urban settings where microclimate variations and the frequency of extreme events are of significant concern (Deng et al., 2023; Shen et al., 2019). Hyper-resolution modeling not only enhances our ability to predict these urban-specific phenomena but also deepens our understanding of the broader impacts of

urbanization on local weather patterns, including temperature, precipitation, and wind dynamics (Li et al., 2021b; Wang and Li, 2019; Liao et al., 2015; He et al., 2019). By capturing the fine-scale variability of urban environments, these models are crucial for developing targeted strategies to mitigate and adapt to climate change at the urban level, ultimately contributing to more resilient and sustainable cities.

The weather research and forecasting (WRF) model is widely used worldwide in the numerical weather prediction and

regional climate modeling communities, known for its high precision, innovative schemes, and comprehensive inclusion of various Earth system processes (Chen et al., 2011). Its applications are becoming increasingly widespread in meteorology and related fields, such as weather services, agriculture, forestry, and renewable energy. Studies have shown that coupling the WRF model with an urban canopy model (WRF/UCM) can improve the simulation of near-surface meteorological elements, effectively enhancing the ability to simulate urban climates (Liao et al., 2014; Shen et al., 2019). However, current

urban models still exhibit significant deficiencies in the accuracy of basic data descriptions and the completeness of key process representations, leading to certain limitations in their application (Best and Grimmond, 2015).

For the urban morphological data required by UCMs, the common approach is to classify urban surfaces based on reference imagery and assign building morphological parameters (such as building height and building ratio) and building characteristic parameters (such as thermal and radiative properties) through lookup tables. For example, the WRF model

classifies urban areas simply into low-density residential areas, high-density residential areas, and industrial/commercial areas (Chen et al., 2011). Similarly, the Community Land Model - Urban (CLMU) model adopts this method but further

refines the classifications by country or region (Jackson et al., 2010; Oleson and Feddema, 2020). Furthermore, Stewart and Oke (2012) introduced the concept of Local Climate Zones (LCZs), which considers three-dimensional building structures and categorizes urban surfaces into 10 types based on factors such as land cover, building structure, materials, and human

activities. This method has also been applied to the WRF model (Demuzere et al., 2023), improving simulation performance to some extent. However, the generation of LCZ datasets depends on expert selection and classification of samples, introducing uncertainties due to variations in remote sensing imagery or sample selection.

With the advancement of remote sensing technology and data generation algorithms, urban data is moving towards higher resolution and greater comprehensiveness, enabling the mapping of high-resolution three-dimensional urban morphological

structures. Urban surface classification data has evolved from an early 1 km resolution with three categories to a 100 m resolution with ten categories (i.e., LCZ datasets) (Demuzere et al., 2022). In addition to developments in the United States, Europe, and mainland China (Li et al., 2020), global high-resolution datasets have also been established (Li et al., 2022; Esch et al., 2022). To meet the needs of UCMs, detailed three-dimensional urban morphological structure datasets have been preliminarily established for some cities in the United States and China (Ching et al., 2009; He et al., 2019; Li et al., 2021a;

Sun et al., 2021). These datasets include building height, building ratio, and frontal area index, providing a good representation of urban three-dimensional morphological structures. The National Urban Database and Access Portal Tool (NUDAPT) provides grid datasets of urban canopy parameters (UCPs) necessary for urban climate modeling systems for 44 city downtown areas in North America (Ching et al., 2009). Additionally, Sun et al. (2021) has shared UCP datasets for 60 cities in China as well. These detailed high-resolution data have begun to be applied in urban simulation studies, showing

certain advantages. For example, Miao et al. (2009) applied detailed UCPs in simulations of the Beijing area, while Dai et al. (2019) used similar detailed UCPs in studies of the Pearl River Delta. They both found that this significantly enhancing the model's simulation capability. In addition, other studies have similarly found that the application of high-resolution UCP datasets leads to varying degrees of improvement in simulation results (Deng et al., 2023; Sun et al., 2021; Shen et al., 2019). Despite this, existing UCP data face challenges in consistency due to differences in data sources and production methods,

making it difficult to form a comprehensive set of input parameters for regional or global urban modeling. More importantly, the currently available UCP datasets are limited to only a few cities and have restricted spatial coverage, making them insufficient for large-scale urban climate simulations. Recently, Kamath et al. (2024) released a global building heights for urban studies (UT-GLOBUS) for city-and-street-scale urban simulations. Although UT-GLOBUS covers more than 1200 cities or locales worldwide, UCP data for East Asia remain unavailable due to the lack of building vector data in this area.

For study areas without detailed UCP data, urban changes can only be described from a two-dimensional perspective, with three-dimensional morphological parameters often represented by a fixed value, failing to reflect the true impact of urban three-dimensional structures on local climates. In addition, Khanh et al. (2023) developed a global 1 km urban morphological dataset by using empirical formulas to estimate UCPs based on gross domestic product (GDP) and population density information. While this dataset performs well in terms of spatial coverage, the accuracy of the estimated parameters

has not yet been compared with results derived from actual building data.

The three-dimensional building footprints can provide essential information for calculating fine-scale UCPs. However, obtaining building-scale footprints with global coverage for calculating detailed global UCPs remain presents a significant challenge currently. Even though global three-dimensional urban height data are becoming more available and their spatial resolution has improved, these high-resolution raster data typically only represent urban heights at a grid scale and do not provide the boundaries and heights of individual buildings, making it difficult to calculate high-resolution UCPs on a global scale. In fact, the OpenStreetMap (OSM) dataset includes vector data for some buildings globally, but coverage is uneven (Herfort et al., 2023). Microsoft offers a global vector dataset, but it lacks building vector data for East Asia and building height data for many regions. However, the latest research has created the first global three-dimensional building footprint dataset (3D-GloBFP) based on publicly available multi-source data (Che et al., 2024). This dataset integrates existing building data to calculate the boundaries and heights of individual buildings globally in 2020. Based on the building vector data, Cheng et al. (2024) developed a global 1 km spatially continuous urban surface property dataset (U-Surf) for the UCM in the Community Earth System Model. However, the urban morphological parameters calculated in U-Surf, including building height, canyon height-to-width ratio, roof fraction, pervious canyon floor fraction, and urban percentage, differ from the UCPs required by WRF/UCM and therefore cannot be directly used in the WRF model. Therefore, the aim of this study is to use a newly-developed building-scale height map to further produce a global spatially continuous high-resolution UCP dataset (hereafter referred to as GloUCP), updating the default parameters in the WRF model to improve simulation accuracy.

## 2 Data and methods

### 2.1 Global building footprint dataset

Vector data that include building outline and height information are essential for computing UCPs. Currently, many studies primarily focus on estimating building heights at the grid scale, often with limited spatial coverage and resolution. This constraint makes it difficult to derive comprehensive global UCPs. Recently, the first global three-dimensional building footprint (3D-GloBFP) dataset was created by leveraging Earth observation data and advanced machine learning techniques (Che et al., 2024). This dataset combines global building boundaries derived from Microsoft's building footprints and the research by Shi et al. (2024), achieving average precisions of over 90% and 80%, respectively, across different regions. Together, these two open-source datasets provide a thorough global spatially continuous building boundary dataset in 2020. To ensure maximum coverage of reference building heights worldwide, the 3D-GloBFP dataset integrates building footprint data with height information from ONEGEO Map, Microsoft building footprints, Baidu Maps, and EMU Analytics (Che et al., 2024). Additionally, they developed height estimation models for 33 global subregions using the extreme gradient boosting (XGBoost) regression method, integrating various remote sensing and building morphology features. The height estimation models demonstrate good performance globally, with $R^2$ values between 0.66 and 0.96, and root mean square errors (RMSEs) ranging from 1.9 m to 14.6 m across the 33 subregions. Overall, the 3D-GloBFP dataset, which can provide

global building 2D footprint polygons along with their heights, is the most comprehensive among existing building vector data, making it a robust foundation for calculating UCPs in this study.

## 2.2 Development of global 1 km spatially continuous UCPs for the WRF model

Urban morphological parameters required by the WRF/UCM model can be calculated using building-scale outline and height data, allowing the derivation of UCPs at any spatial resolution. These parameters include mean building height, standard deviation of building height, area weighted mean building height, plan area fraction, building surface to plan area ratio, frontal area index, and distribution of building heights, as detailed in Table 1. They can be applied to three types of UCMs in the WRF model: single-layer urban canopy model (SLUCM), building effect parameterization (BEP), and BEP-

BEM (building energy model). In this study, all the UCPs are developed globally at a resolution of approximately 1 km (i.e., 1/120°) based on the building-scale information (i.e., building outline and height) provided by the 3D-GloBFP dataset. Additionally, to ensure the consistency of the calculation area with the existing impervious surface extent, we further use the Global Artificial Impervious Area (GAIA) dataset in 2020 as a mask for UCP calculation. The GAIA dataset is generated based on long-term optical remote sensing data from the Landsat series of satellites, supplemented by VIIRS nighttime light

data and Sentinel-1 radar data (Gong et al., 2020). It uses spatial masking and feature evaluation algorithms to achieve rapid mapping of impervious surfaces, and employs a time consistency verification algorithm to filter and infer logical sequences of impervious surfaces, ensuring their spatial and temporal rationality. In our dataset, only grids with an impervious surface ratio exceeding 1% are retained. Moreover, we have provided 1 km resolution impervious surface fraction data for urban areas in 2020 derived from the GAIA dataset as well. This allows users to conveniently define urban categories (i.e., low-

density residential, high-density residential, and industrial/commercial) in WRF simulations based on the consistent impervious fraction data. Once the urban type of each grid is determined, our dataset can be used to reassign urban morphological parameters for each grid, thereby providing a more detailed and accurate depiction of urban morphological variations within the study areas.

## 2.3 Comparison between new and existing UCPs for the WRF model

To demonstrate the reliability of the GloUCP dataset generated in this study, we select three major urban agglomerations in China (i.e., Beijing-Tianjin-Hebei region, Yangtze River Delta, and Guangdong-Hong Kong-Macao Greater Bay Area) and three important cities in the United States (i.e., Seattle, San Francisco, and Philadelphia) as representatives. This selection is based on the availability of data and the representativeness of their geographical distribution. We systematically evaluate the consistency between reference data, our new dataset, and comparison datasets using the coefficient of determination ($R^2$) and

RMSE as statistical indicators.

**Table 1. Calculation of GloUCP for the WRF model and the applied UCP schemes.**

| Variable | Abbreviation | Formula | Description | Used by UCM (URB_PARAM Index) |
|---|---|---|---|---|
| Mean building height | $\bar{h}$ | $$\bar{h} = \frac{1}{N}\sum_{i=1}^{N} h_i$$ | $h_i$ is the height of building $i$; $N$ is the total number of buildings in the grid; | SLUCM (92) |
| Standard deviation of building height | $h_{std}$ | $$h_{std} = \sqrt{\frac{\sum_{i=1}^{N}(h_i - \bar{h})}{N-1}}$$ | | SLUCM (93) |
| Area weighted mean building height | $h_{aw}$ | $$h_{aw} = \frac{\sum_{i=1}^{N} A_i h_i}{\sum_{i=1}^{N} A_i}$$ | $A_i$ is the plan area on the ground level of building $i$; | SLUCM, BEP, BEP-BEM (94) |
| Plan area fraction | $\lambda_p$ | $$\lambda_p = \frac{A_p}{A_T}$$ | $A_p$ is the total footprint area of buildings in the grid; $A_T$ is the total area of the grid; | SLUCM, BEP, BEP-BEM (91) |
| Building surface to plan area ratio | $\lambda_b$ | $$\lambda_p = \frac{A_R + A_W}{A_T}$$ | $A_R$ is the total roof area of buildings in the grid; $A_W$ is the total area of non-horizontal roughness elements (such as walls); | SLUCM, BEP, BEP-BEM (95) |
| Frontal area index | $\lambda_f$ | $$\lambda_f(\theta) = \frac{A_{proj}}{A_T}$$ | $A_{proj}$ is the total projected area of buildings on a plane perpendicular to four wind directions (0°, 135°, 45°, 90°,); $\theta$ is the wind direction. | SLUCM (96-99) |
| Distribution of building heights | $h_{dis}(i)$ | $$h_{dis}(i) = \frac{N_{dis}(i)}{N} \times 100\%$$ | $N_{dis}(i)$ is the number of buildings vertically resolved with 5 m bins spanning 0-75 m. | BEP, BEP-BEM (118-132) |

Notes: UCM, urban canopy model; SLUCM, single-layer urban canopy model; BEP, building effect parameterization; BEM, building energy model. The values in parentheses in the last column represent the index of the UCP in the URB_PARAM array.

For China, we use building height data obtained from the Baidu Maps API (https://ditu.baidu.com) as the reference data, and the UCP dataset released by Sun et al. (2021) (hereafter referred to as Sun2021) as the comparison dataset. Both the GloUCP and Sun2021 have a spatial resolution of 1 km, while the Baidu Maps data is at the building scale. Considering the differences in resolution among these datasets, we process the building-scale Baidu Maps data in the same method as the initial building height estimation dataset, resulting in 1 km resolution urban height data from Baidu Maps. We then conduct consistency analysis using the pixel values within the spatial extent where all three datasets overlap.

For the United States, we use a building footprint dataset with height information released by Microsoft in 2017 (https://wiki.openstreetmap.org/wiki/Microsoft_Building_Footprint_Data#March_2017_Release) as the reference data and the NUDAPT dataset, which includes UCPs for 44 cities in the United States developed from airborne LiDAR data (Ching et al., 2009), as the comparison dataset. Both our GloUCP dataset and NUDAPT dataset have a spatial resolution of 1 km, while the Microsoft data is at the building scale. Similar to the analysis in China, we process the building-scale Microsoft data to obtain 1 km resolution urban height data and conduct consistency analysis using the overlapping spatial extent of all three datasets.

To demonstrate the advantages of our dataset, we further compared it with the recently released UT-GLOBUS dataset by Kamath et al. (2024) and the global urban morphological dataset developed by Khanh et al. (2023). Additionally, we compare the spatial distribution of the default UCPs for low-density residential areas, high-density residential areas, and industrial/commercial areas defined in the current WRF model with our GloUCP dataset. This comparison aims to assess not only the heterogeneity in their geographical distribution but also the differences in their numerical characteristics. This will provide a basis for further exploring the feasibility of using the new dataset in WRF simulations to enhance urban climate modeling performance.

## 3 Results and discussion

### 3.1 Global distribution of the GloUCP

Fig. 1 illustrates the spatial distribution of mean building height across global land. Overall, in economically developed and highly urbanized regions, such as the eastern coast of the United States, Western Europe, Japan, and eastern China, the mean building height is relatively high. Conversely, in most parts of Africa and South America, the mean building height is much lower. The area-weighted mean building height follows a similar spatial pattern to that of the mean building height, being higher in regions with advanced urbanization and lower in areas where urbanization is less developed.

We further examine the spatial distribution patterns of mean building height within the three study regions: China, the contiguous United States, and Europe. In China, the mean building height generally follows a pattern of being higher in the east than in the west, and higher in coastal areas than in inland regions. For instance, the mean building heights in some eastern coastal cities, such as Shanghai, Beijing, and Tianjin, respectively, which are significantly higher than those in inland areas. At the provincial level, most provinces in China are dominated by low-rise buildings, while the proportion of multi-

rise buildings is higher in the eastern region, and high-rise buildings are more prevalent in Hong Kong and Macau (Fig. 1b). Focusing on the three major urban agglomerations in China, the Yangtze River Delta (YRD) stands out with a larger scale of multi-rise and high-rise buildings compared to the other two regions. The mean building height in the YRD is 10.62 m, higher than the 9.93 m in the Guangdong-Hong Kong-Macao Greater Bay Area (GBA) and the 8.24 m in the Beijing-Tianjin-Hebei (BTH) region (Figs. 2b1-2d1).

In the United States, building height distribution follows a general pattern of decreasing heights from coastal areas to inland regions. The vast agricultural states in the Midwest and western regions have low mean building heights, mostly below 10 m, while the northeastern states, as well as California and Florida, have higher mean building heights (Figs. 3b1-3d1). At the state level, apart from Washington, D.C., the proportion of high-rise buildings in other states is very small, with the proportion of multi-rise buildings decreasing from the coasts to the interior (Fig. 1c). In Europe, the proportion of high-rise buildings is generally low across different countries (Fig. 1d). The mean building height in European countries is about 6.81 m, lower than the 8.75 m observed in the United States and the 8.33 m in China. Moreover, the proportion of high-rise building area in the city centers of European regions is also less compared to both the United States and China (Figs. 4b1-4d1).

The standard deviation of building height reflects the spatial heterogeneity of building distributions and is often used to indicate surface roughness. Overall, the standard deviation of building height is larger in major cities in Europe, the eastern coastal areas of China, Japan and South Korea (Fig. S1b). In China, the standard deviation of building height is about 2.91 m. Its spatial distribution exhibits a pattern where southern regions have higher values than northern regions, and coastal areas have higher values than inland areas. The YRD (5.71 m) and GBA (6.20 m) have similar standard deviation values, both significantly higher than that of the BTH region (3.10 m) (Figs. 2b2-2d2). In the United States, apart from some northeastern states, the standard deviation is generally low, with values below 1.5 m in most areas. Specifically, Seattle has a standard deviation of 2.38 m, San Francisco 5.10 m, and Philadelphia 3.48 m (Figs. 3b2-3d2). In suburban areas, buildings are generally low and flat, with significant building height variations occurring only in certain parts of city centers. In Europe, the standard deviation of building height is about 2.34 m and is relatively uniform across regions, except in a few countries. For instance, Paris has a notably higher standard deviation of 10.84 m, which is significantly larger than those in London (3.42 m) and Berlin (4.87 m) (Figs. 4b2-4d2). Overall, the spatial distribution of height standard deviation exhibits a certain similarity to that of mean building height. Regions with higher mean building heights also tend to have greater height standard deviations.

The plan area fraction and building surface to plan area ratio help to understand building density and land use efficiency. These indicators show higher values in major cities in North America, Western Europe, the eastern coastal areas of China, Japan, and Southeast Asia, where high urbanization levels lead to extensive surface coverage by tall and densely clustered buildings (Fig. S2). The spatial distribution of the frontal area index generally mirrors that of the plan area fraction (Fig. S3). Overall, this pattern suggests that building density is higher in Europe, followed by the United States and China.

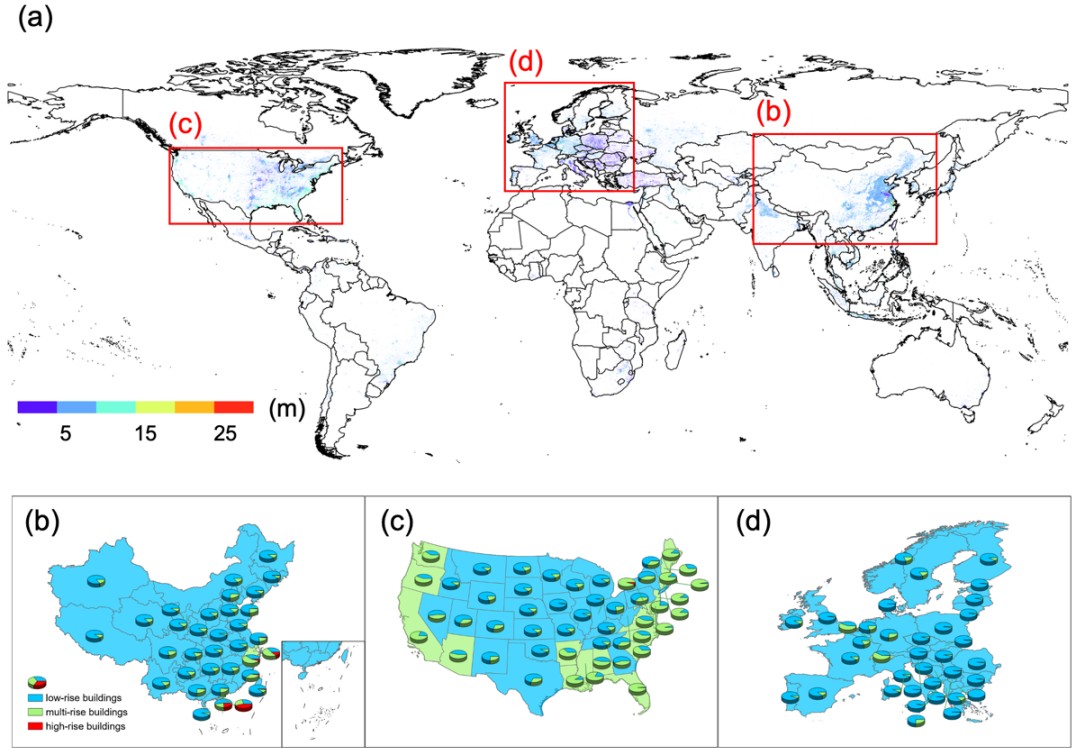

**Figure 1. The spatial pattern of mean building height across global land.** (b-d) show the spatial distribution of different building type proportions in China, the contiguous United States, and Europe, as highlighted in (a). Specifically, Low-rise buildings are defined as those with heights less than 10 m, multi-rise buildings as those between 10 and 24 m, and high-rise buildings as those exceeding 24 m. The pie charts represent the proportion of these three building types in various subregions,

with the color of each subregion indicating the predominant building type in that area.

### 3.2 Comparison with existing UCP products for the WRF model

In China, we used Baidu's building height data as reference data. By comparing this with the Sun2021 dataset, we found that our GloUCP dataset significantly outperforms in terms of spatial continuity and coverage. GloUCP effectively fills in gaps in

existing datasets, particularly for buildings in suburban areas of large cities, as well as in small to medium-sized cities and rural areas (Fig. 5). This comprehensive spatial coverage is crucial for regional climate modeling; without it, WRF/UCM would rely on lookup tables to fill in missing UCP values for areas not covered by data. This could lead to inconsistencies in UCPs across the simulation domain, potentially compromising the accuracy of the simulation results.

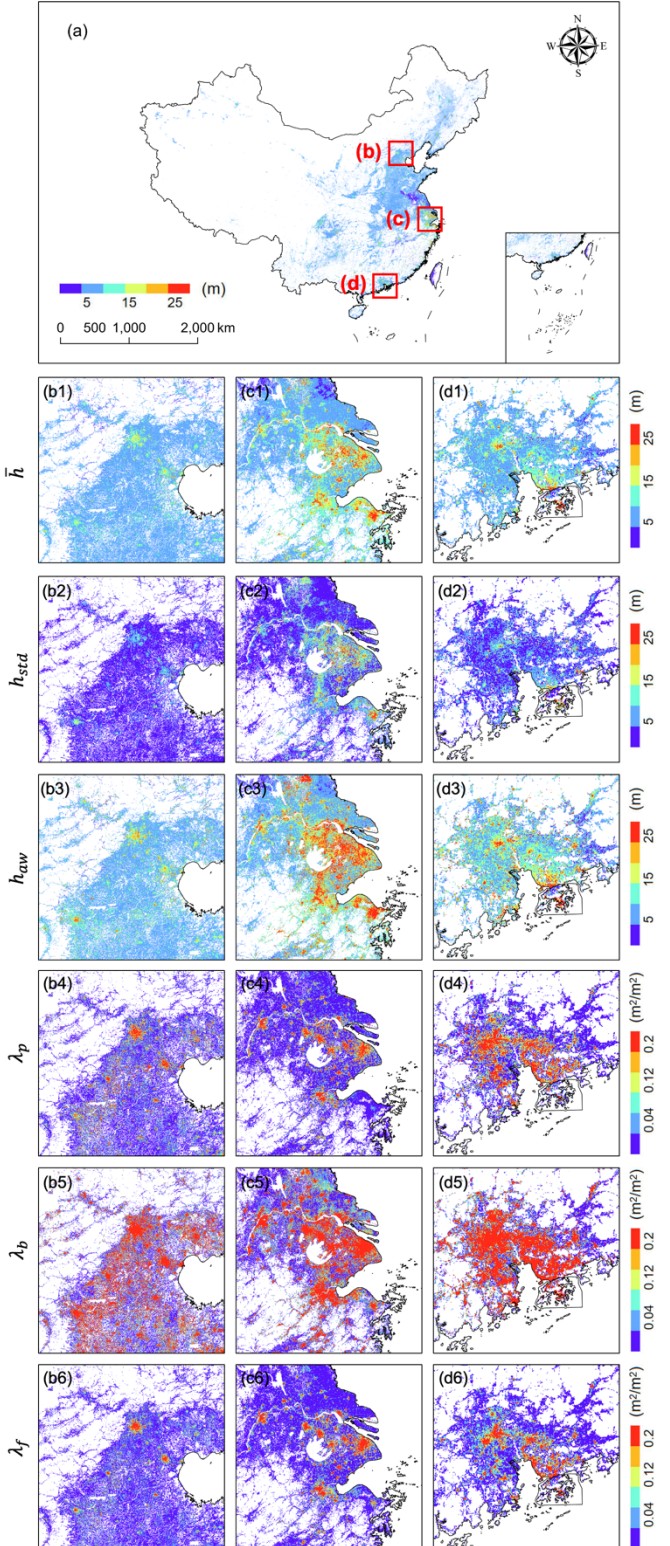

**Figure 2. The spatial distribution of UCPs in China.**
(b-d) show the spatial distribution of mean building height, standard deviation of building height, area weighted mean building height, plan area fraction, building surface to plan area ratio, and frontal area index for three major urban agglomerations, i.e., Beijing-Tianjin-Hebei region, Yangtze River Delta, and Guangdong-Hong Kong-Macao Greater Bay Area, as highlighted in (a).

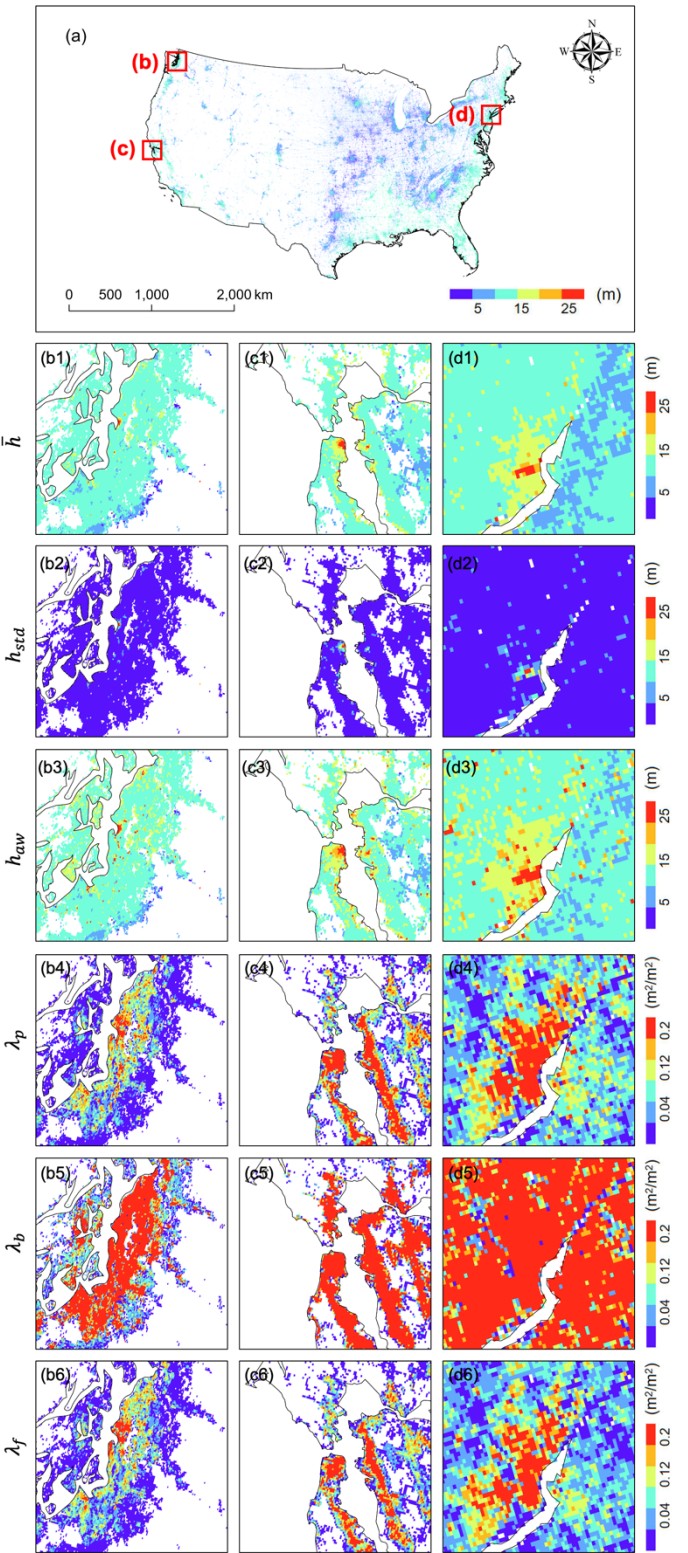

**Figure 3. The spatial distribution of UCPs in the contiguous United States.** (b-d) show the spatial distribution of mean building height, standard deviation of building height, area weighted mean building height, plan area fraction, building surface to plan area ratio, and frontal area index for three major cities, i.e., Seattle, San Francisco, and Philadelphia, as highlighted in (a).

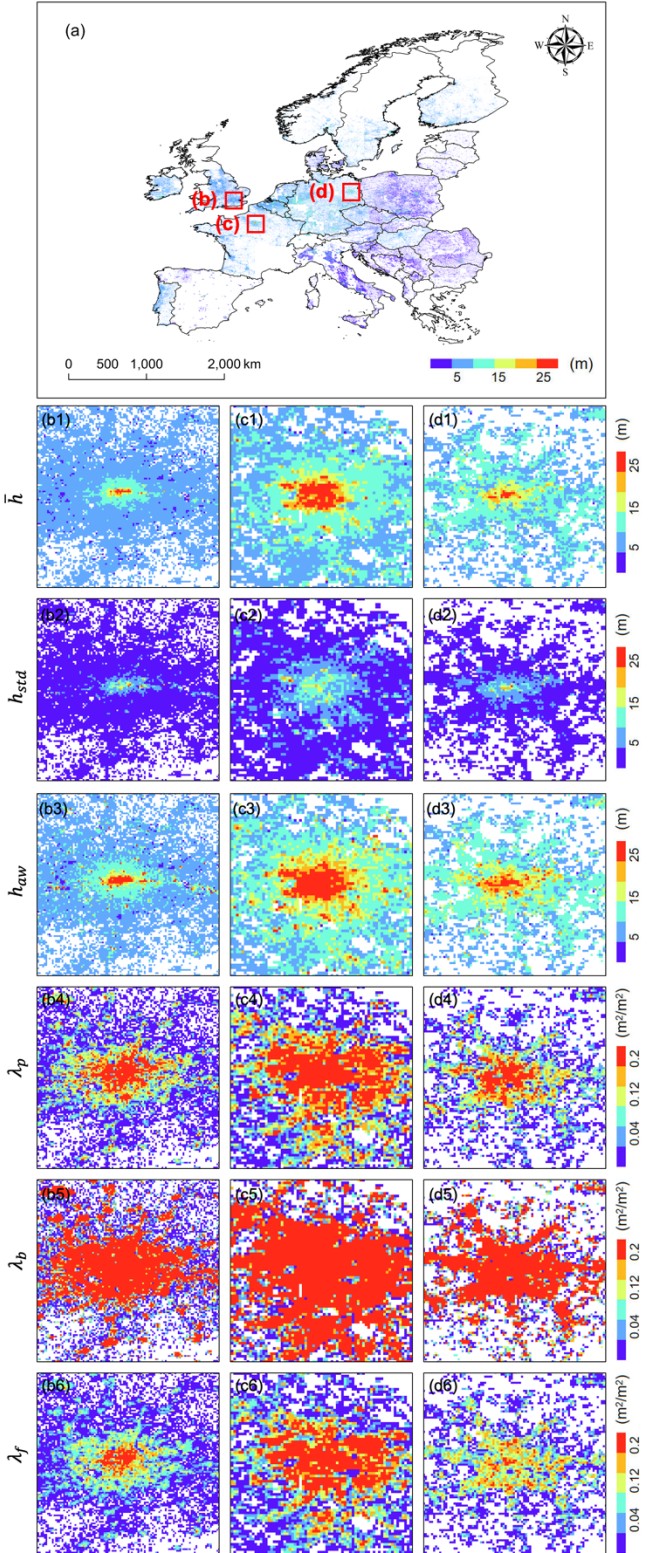

**Figure 4. The spatial distribution of UCPs in Europe.**
(b-d) show the spatial distribution of mean building height, standard deviation of building height, area weighted mean building height, plan area fraction, building surface to plan area ratio, and frontal area index for three major cities, i.e., London, Paris, and Berlin, as highlighted in (a).

Fig. 6 shows a pixel-scale comparison of mean building heights in GloUCP, reference data, and Sun2021 across three major urban agglomerations in China. In these three regions, the coefficient of determination ($R^2$) for GloUCP is 0.19, 0.20, and 0.17 in BTH, YRD, and GBA regions, respectively, higher than the 0.09, 0.06, and 0.07 for Sun2021 (Table S1). This indicates that although the accuracy is relatively low, GloUCP more accurately reflects the true distribution of building heights in these areas. From the perspective of RMSE, GloUCP consistently outperforms Sun2021 across all three regions, with lower RMSE values, indicating higher accuracy in building height predictions. In the BTH region, GloUCP has an RMSE of 14.32 m, lower than Sun2021's 15.22 m; in the YRD, GloUCP's RMSE is 15.88 m, well below Sun2021's 16.79 m; and in the GBA, GloUCP's RMSE is 17.88 m, also better than Sun2021's 19.29 m (Table S1). Furthermore, when RMSE is compared across different building height intervals (i.e., ≤10 m, 10-24 m, and >24 m), GloUCP generally shows lower RMSE values than Sun2021 in nearly all height categories, with a particularly noticeable advantage in lower buildings. This also indicates that, for both our GloUCP dataset and the Sun2021 dataset, the errors in mean building height primarily stem from taller buildings. Nevertheless, GloUCP has a significant advantage in predicting building height data across China, especially in complex urban areas and for high-rise buildings. Overall, GloUCP not only better reflects the actual distribution of building heights but also exhibits superior spatial coverage across different geographical regions.

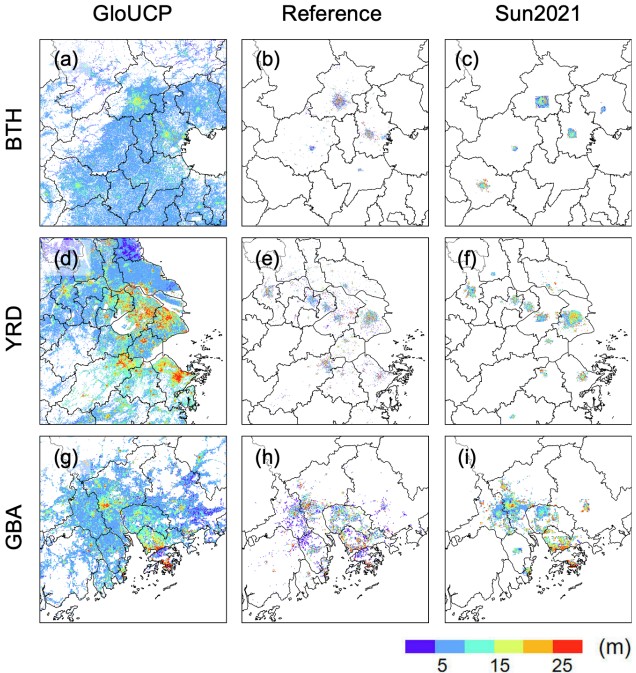

**Figure 5. Comparison of the spatial distribution of mean building heights in GloUCP, reference data, and Sun2021 across three major urban agglomerations in China.**

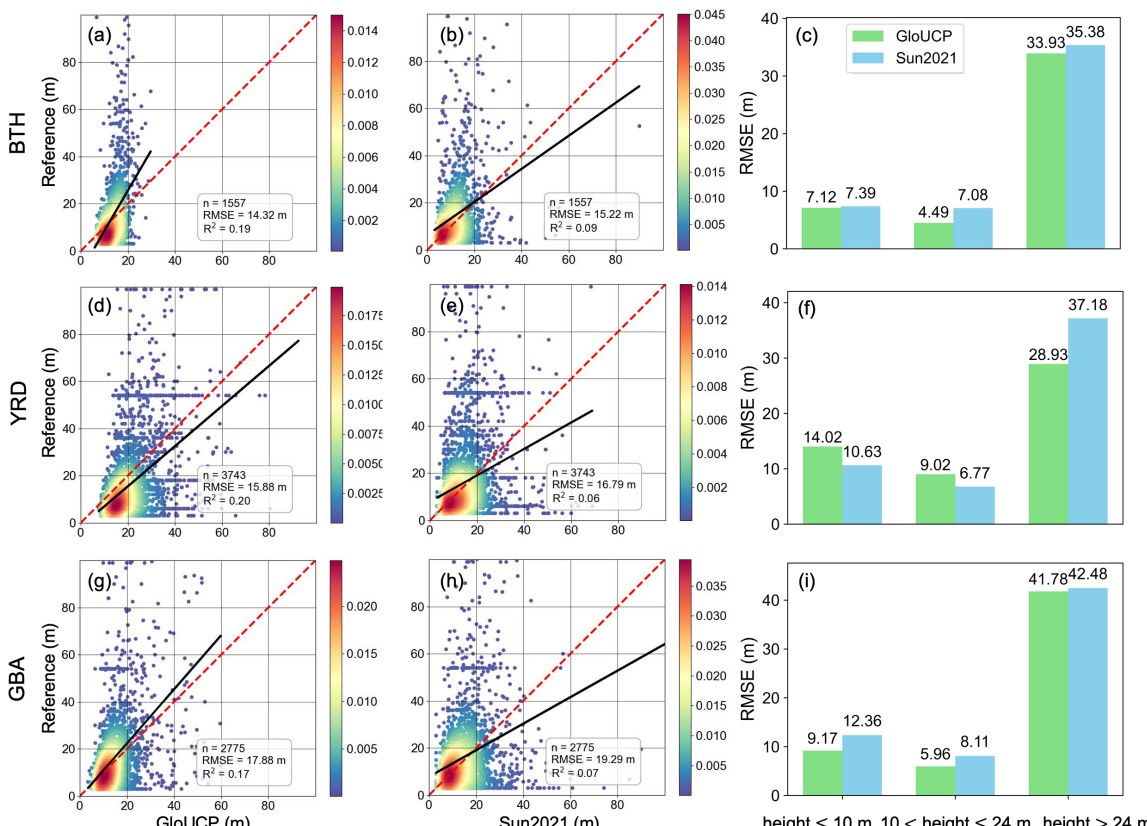

**Figure 6. Pixel-scale comparison of mean building heights in GloUCP, reference data, and Sun2021 across three major urban agglomerations in China.** The red dashed line represents the 1:1 line, while the black solid line indicates the fitted regression line.

For the results in the United States, we used a building footprint dataset with height information released by Microsoft in 2017 as the reference data. Comparing this with the NUDAPT dataset, we found that GloUCP demonstrates a more comprehensive distribution of building heights across the three cities. GloUCP's coverage is more extensive, capturing a wider area and performing well in both lower and higher building heights, such as in the southern region of Philadelphia (Fig. 7). In contrast, the NUDAPT data is primarily concentrated in the city centers, with a more limited and concentrated height distribution that fails to cover a broader area. This limitation is particularly evident in Seattle and Philadelphia, where NUDAPT's spatial coverage is restricted.

In terms of building height consistency, GloUCP's $R^2$ values in Seattle, San Francisco, and Philadelphia are 0.81, 0.83, and 0.52, respectively, which are comparable to or better than NUDAPT's values of 0.64, 0.73, and 0.39 (Table S2). When looking at RMSE, GloUCP's values for Seattle, San Francisco, and Philadelphia are 2.51 m, 4.73 m, and 5.50 m, respectively, lower than NUDAPT's 9.03 m, 8.57 m, and 8.50 m. This indicates better consistency between GloUCP and the

reference data. Furthermore, across different building height intervals in all three cities, GloUCP consistently shows lower
RMSE values than NUDAPT. This highlights GloUCP's superior performance in predicting building heights. Overall,
GloUCP outperforms NUDAPT in the three U.S. cities, particularly in terms of spatial coverage and prediction accuracy of
building heights. GloUCP is better at capturing variations in building heights both within and around urban areas, and its
exceptional performance makes it highly valuable for urban modeling and climate simulations.

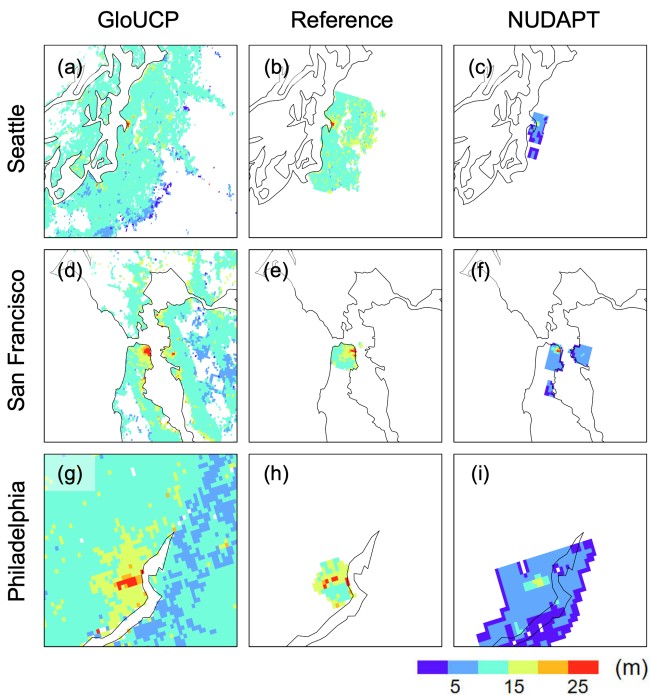

**Figure 7. Comparison of the spatial distribution of mean building heights in GloUCP, reference data and NUDAPT across three representative cities in the United States.**

Nevertheless, using Microsoft's data as reference does not necessarily imply that its height values are absolutely accurate.
The heights in Microsoft's dataset were interpolated using a digital terrain model derived from very high-resolution aerial
photography, with building boundaries that were hand-digitized. NUDAPT's data were derived from LiDAR measurements,
which are also highly accurate. However, it is important to note that the NUDAPT dataset was created using data from the
year around 2009, whereas our dataset is based on data from around 2020, leading to a temporal discrepancy. As a result,
there is some degree of uncertainty in these comparisons. When directly comparing the mean building heights between
GloUCP and NUDAPT in these three cities, we found that GloUCP generally shows higher mean building heights than
NUDAPT (Fig. S4). Additionally, the $R^2$ values between GloUCP and NUDAPT for Seattle, San Francisco, and

Philadelphia are 0.63, 0.87, and 0.70, respectively, indicating a strong level of consistency between our dataset and the NUDAPT data as well.

Furthermore, we compared our dataset with the recently released UT-GLOBUS dataset by Kamath et al. (2024) and the global urban morphological dataset developed by Khanh et al. (2023). Because the UT-GLOBUS dataset only provides three parameters (i.e., area weighted mean building height, plan area fraction, and building surface to plan area ratio), we conducted a comparison of these parameters in three cities in the United States (Figs. S5-S7). Overall, our GloUCP dataset and the UT-GLOBUS dataset exhibit similar levels of accuracy. Compared to the reference data, both datasets show relatively high estimation accuracy for most parameters, except for an underestimation of the building surface to plan area ratio. On average, the $R^2$ values for our GloUCP dataset across the three cities are generally above 0.8, slightly outperforming the UT-GLOBUS dataset. Nevertheless, our dataset offers more comprehensive spatial coverage of global urban areas, particularly in East Asia. This broader coverage provides greater support for urban climate simulations, especially for small and medium-sized cities worldwide.

For the global urban morphological dataset developed by Khanh et al. (2023), we used their UCPs estimated based on GDP and population density information in 2010 (hereafter referred to as Knanh2010) to compare it with our data from the representative regions in China and the United States, respectively. Generally, the spatial coverage of our GloUCP dataset is still larger than that of Knanh2010 (Figs. S8 and S9). In China, compared to the reference data, the Knanh2010 dataset performs well in capturing low- to mid-rise buildings but significantly underestimates the height of high-rise buildings (Fig. S10). In the United States, the accuracy of our GloUCP dataset is similar to that of Knanh2010, though our dataset performs slightly better across different building height categories (Fig. S11). Overall, while the Knanh2010 dataset already offers better spatial coverage than most existing datasets, our GloUCP dataset provides even greater coverage. In China, the accuracy of most datasets remains suboptimal, but our dataset slightly outperforms others, particularly in representing high-rise buildings.

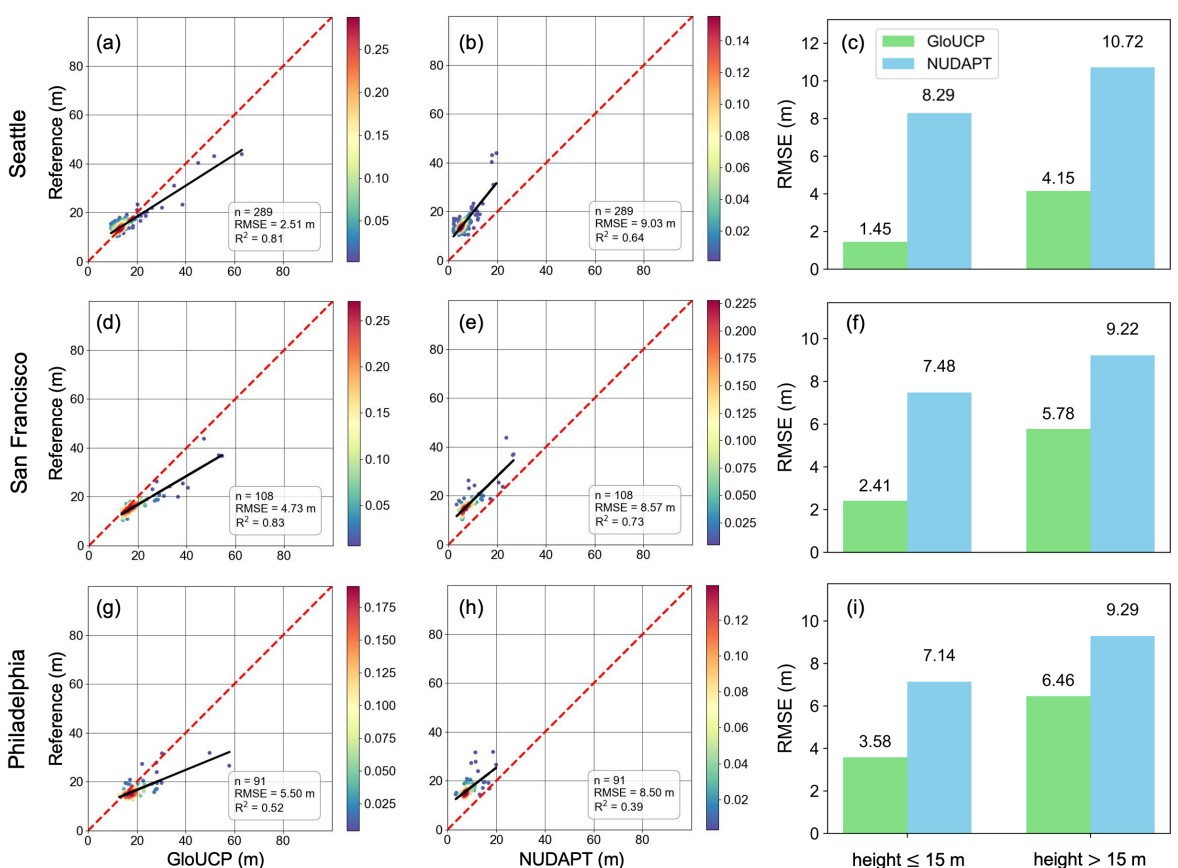

**Figure 8. Pixel-scale comparison of mean building heights in GloUCP, reference data and NUDAPT across three representative cities in the United States.** The red dashed line represents the 1:1 line, while the black solid line indicates the fitted regression line.

### 3.3 Comparison with the default UCPs in the WRF model

To thoroughly assess the applicability of the dataset constructed in this study for WRF simulations, we compared it with the default UCPs currently widely used in the WRF model. To reflect the impact of urban three-dimensional structures on meteorological processes, the prevailing approach in WRF simulations is to further subdivide urban land cover into three categories, i.e., low-density residential areas, high-density residential areas, and commercial areas, each assigned a fixed UCP value. Specifically, low-density residential areas have an impervious surface ratio of less than 50%, corresponding to a

building height of 5 m; high-density residential areas have an impervious surface ratio between 50% and 80%, with a building height of 7.5 m; and industrial/commercial areas have an impervious surface ratio greater than 80%, with a building height of 10 m. We extracted the default mean building height data from the WRF model and compared it with our GloUCP dataset to analyze the differences in data characteristics and spatial distribution.

Fig. 9 compares the default mean building height in the WRF model with its distribution in GloUCP. Overall, the default
height significantly underestimates building heights in various urban regions. Whether in the three major urban
agglomerations in China or the three representative cities in the United States, many buildings in city centers reach heights of
10 m or even over 20 m, which the default data fails to capture, particularly in the case of high-rise buildings.

Figs. 10 and 11 illustrate the spatial distribution of mean building heights in GloUCP and the default values in the WRF
model across three major urban agglomerations in China and three representative cities in the United States, respectively.
From a spatial distribution perspective, the default dataset can to some extent reflect the higher building heights in city center
areas, but its height values are generally lower than those calculated by the GloUCP dataset. Moreover, GloUCP exhibits
significantly greater spatial heterogeneity, providing a more detailed and accurate depiction of building height variations
within the study areas.

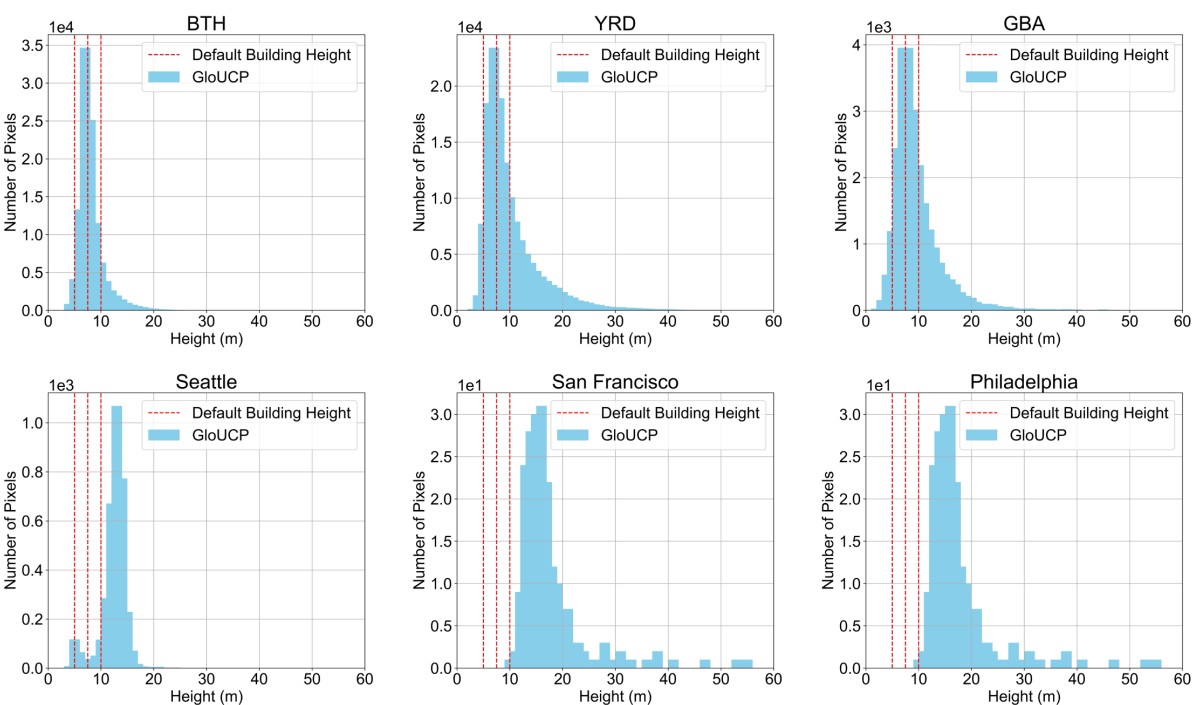


**Figure 9. Distribution of mean building height in three major urban agglomerations in China and three
representative cities in the United States.** The red dashed lines denote low-density residential areas with a default building
height of 5 m, high-density residential areas with a default building height of 7.5 m, and industrial/commercial areas with a
default building height of 10 m, respectively.


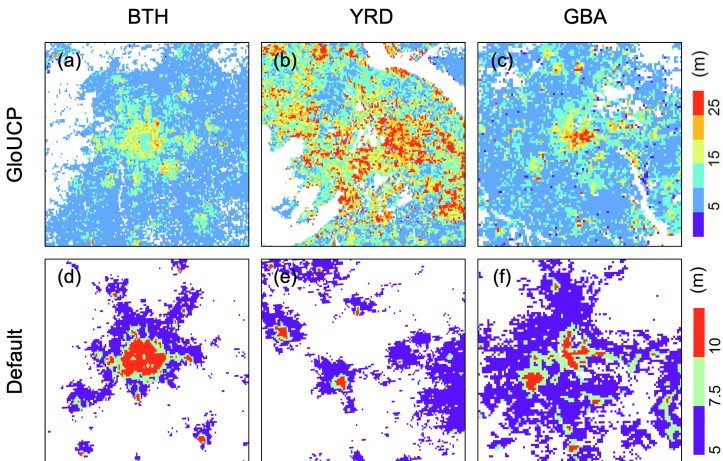

**Figure 10. Comparison of the spatial distribution of mean building heights in GloUCP and default values in the WRF model across three major urban agglomerations in China.**

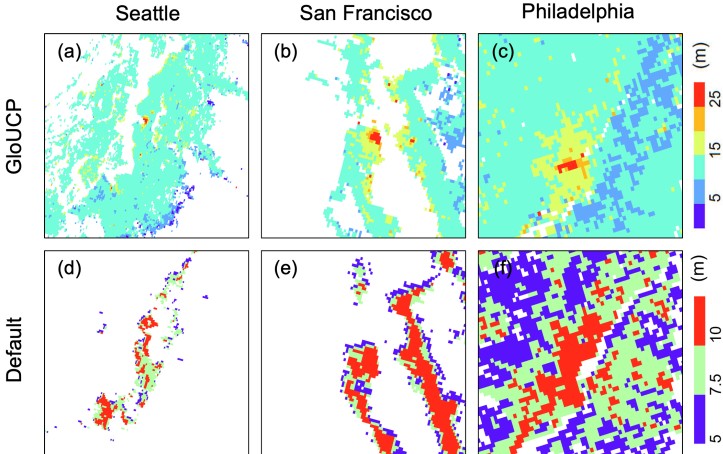

**Figure 11. Comparison of the spatial distribution of mean building heights in GloUCP and default values in the WRF model across three representative cities in the United States.**

## 4 Limitations and uncertainty

The primary purpose of GloUCP is to provide a global 1 km spatially continuous UCPs for three types of UCMs (i.e., SLUCM, BEP, and BEP-BEM) in the WRF model. The uncertainties in GloUCP data primarily originate from the 3D-GloBFP dataset, which is generated from multi-source datasets that integrate information with varying spatiotemporal coverage. Additionally, in regions with limited building height samples (e.g., Africa), the accuracy of building height

estimation remains relatively low. Therefore, as the accuracy of the 3D-GloBFP dataset improves, the precision of the GloUCP dataset can also be further enhanced.

Considering computational and storage costs, this study only provides UCPs at a global scale with a 1 km resolution. However, based on building vector data, UCP datasets at any spatial resolution can be generated. When regional climate simulations require higher spatial resolution, the resolution of UCP calculations can be adjusted to meet the needs of high-resolution climate modeling. It is important to acknowledge these limitations and uncertainties when using GloUCP data for modeling and analysis. Despite these limitations, GloUCP provides globally comprehensive urban canopy parameters,

supporting detailed urban climate simulations on a global scale.

## 5 Data availability

The 1 km GloUCP dataset which is stored as WRF binary file format is publicly available at figshare: https://doi.org/10.6084/m9.figshare.27011491 (Liao et al., 2024).

## 6 Conclusions

UCPs play a critical role in urban climate modeling, as they significantly influence the accuracy of simulations that are essential for understanding the impacts of urbanization on local and regional climates. Despite the importance of UCPs, publicly available datasets for the WRF model are currently limited, covering only 44 cities in the United States and 60 in China. Although several global UCP datasets have been released in recent years, they still have limitations in terms of spatial coverage and accuracy. These limitations underscore the need for more comprehensive and globally applicable UCP datasets.

In this study, we developed a global 1 km spatially continuous UCP dataset — GloUCP, utilizing the latest available building-level information in 2020. It can be applied to all three types of UCMs (i.e., SLUCM, BEP, and BEP-BEM) in the WRF model. The GloUCP dataset has proven to be highly effective and accurate in capturing UCPs across various regions, including highly urbanized areas in China and key metropolitan areas in the United States. Through extensive comparisons with existing datasets, such as Sun2021 in China, NUDAPT in the United States, the UT-GLOBUS dataset developed by

Kamath et al. (2024), and the global UCP dataset developed by Khanh et al. (2023), GloUCP has demonstrated superior spatial coverage and improved precision in predicting building heights. These attributes make GloUCP a comprehensive and reliable dataset for global urban canopy parameterization, offering significant advancements over existing datasets.

The extensive coverage and high-resolution data provided by GloUCP are invaluable for researchers and urban planners aiming to enhance the accuracy of urban climate simulations. Such improvements are crucial for better understanding the

impacts of urbanization on local and regional climates. Previous studies have already confirmed that using accurate UCP parameters can enhance the precision of urban climate simulations. However, the primary objective of this study was not to quantify the extent to which fine-scale and spatially complete UCPs improve simulation accuracy through case studies.

Instead, our goal was to provide a globally complete and high-resolution UCP dataset that can serve as a foundational tool for future urban climate modeling research. We hope that subsequent studies will further explore the potential of this dataset

to enhance urban climate simulations and contribute to more informed decision-making in urban planning and climate mitigation efforts.

**Author contributions.** W. Liao and X. Liu designed the research. Y. Li, Y. Wang, and Y. Che performed the experiments and organized the dataset. W. Liao and Y. Li wrote the original manuscript. All authors reviewed and revised the manuscript.


**Competing interests.** The authors have no competing interests to declare.

**Acknowledgements.** We gratefully acknowledge the creation and provision to the Building Footprint and Height dataset by Microsoft and Baidu Map.


**Financial support.** This study was supported by the National Natural Science Foundation of China (grants U2342227 and 42271419), National Science Fund for Distinguished Young Scholars (grant 42225107), Fundamental Research Funds for the Central Universities, Sun Yat-sen University (grant 23lgbj014), Young Talent Support Project of Guangzhou Association for Science and Technology (grant QT-2023-010), and Science and Technology Projects in Guangzhou (grant
2023A04J1515).

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
