# Peer review of "GloUCP: A global 1 km spatially continuous urban canopy parameters for the WRF model"

_Earth System Science Data, 2024_

## Author Comment (AC1)

We thank the reviewer for your constructive comments, which significantly improve the quality of our manuscript. Based on these comments, we have revised the manuscript thoroughly. The changes made to the manuscript are noted in the revised manuscript, and described in detail below (reviewer comments are in *italic* and cited texts are in **bold**).

*The authors used an existing global three-dimensional vector data of buildings (3D-GloBFP) to compile a global 1 km spatially continuous UCP (GloUCP) dataset for application in the WRF model. They found that GloUCP not only surpasses existing datasets in accuracy but also provides superior spatial coverage. This dataset would be very useful for future improvements of WRF-Urban modeling. Before it can be considered for publication, I have a few concerns and suggestions for the authors to consider.*

**Response**: We sincerely appreciate the reviewer's positive feedback on our research. Based on your insightful comments, we have thoroughly revised the manuscript. We hope our responses and revisions meet your expectations.

*Major comments:*

*My major concern is that there are several recent developments of global urban parameter datasets (e.g., UT-GLOBUS data (Kamath et al., 2024: https://doi.org/10.1038/s41597-024-03719-w); U-SURF data (Cheng et al., 2024: https://doi.org/10.5194/essd-2024-416)), which however are totally missing in the introduction and discussion parts of this study. More importantly, what are the novelty, advantages, and uniqueness of GloUCP compared with those recent urban parameter datasets? This needs to be clarified. It would also be useful to conduct comparisons between the GloUCP parameters with those other recent global datasets.*

**Response**: We sincerely appreciate the reviewer for this comment. In fact, both our GloUCP dataset and the U-Surf dataset developed by Cheng et al. (2024) are derived from the same building vector data, i.e., 3D-GloBFP developed by Che et al. (2024), to calculate urban morphological parameters. Theoretically, there are no differences in the mean building height between the two datasets. However, the urban morphological parameters provided by U-Surf, which are designed for the UCM in the

Community Earth System Model, differ from the UCPs required by WRF/UCM and therefore cannot be directly applied in the WRF model. Therefore, the uniqueness of our study is to use a newly-developed building-scale height map to further produce a global, spatially continuous, high-resolution UCP dataset specifically tailored for the WRF model.

Following the reviewer's suggestion, we compare our GloUCP dataset with the UT-GLOBUS dataset developed by Kamath et al. (2024) in three cities in the United States. Overall, our dataset has good consistency with the UT-GLOBUS dataset in terms of accuracy (Figs. S5-S7). However, the key advantage of our dataset is its comprehensive coverage of global urban areas. In contrast, the UT-GLOBUS dataset has significant data gaps, particularly in regions such as East Asia.

We have added a brief description to both datasets in the revised manuscript's introduction section and highlight the advantages of GloUCP compared with those recent UCP datasets. Additionally, we included a comparison between our dataset and the UT-GLOBUS dataset in the results section to highlight their differences, as detailed below.

**"Recently, Kamath et al. (2024) released a global building heights for urban studies (UT-GLOBUS) for city-and-street-scale urban simulations. Although UT-GLOBUS covers more than 1200 cities or locales worldwide, UCP data for East Asia remain unavailable due to the lack of building vector data in this area."**

**"Based on the building vector data, Cheng et al. (2024) developed a global 1 km spatially continuous urban surface property dataset (U-Surf) for the UCM in the Community Earth System Model. However, the urban morphological parameters calculated in U-Surf, including building height, canyon height-to-width ratio, roof fraction, pervious canyon floor fraction, and urban percentage, differ from the UCPs required by WRF/UCM and therefore cannot be directly used in the WRF model. Therefore, the aim of this study is to use a newly-developed building-scale height map to further produce a global spatially continuous high-resolution UCP dataset (hereafter referred to as GloUCP), updating the default parameters in the WRF model to improve simulation accuracy."**

**"Furthermore, we compared our dataset with the recently released UT-GLOBUS dataset by**

**Kamath et al. (2024) and the global urban morphological dataset developed by Khanh et al. (2023). Because the UT-GLOBUS dataset only provides three parameters (i.e., area weighted mean building height, plan area fraction, and building surface to plan area ratio), we conducted a comparison of these parameters in three cities in the United States (Figs. S5-S7). Overall, our GloUCP dataset and the UT-GLOBUS dataset exhibit similar levels of accuracy. Compared to the reference data, both datasets show relatively high estimation accuracy for most parameters, except for an underestimation of the building surface to plan area ratio. On average, the $R^2$ values for our GloUCP dataset across the three cities are generally above 0.8, slightly outperforming the UT-GLOBUS dataset. Nevertheless, our dataset offers more comprehensive spatial coverage of global urban areas, particularly in East Asia. This broader coverage provides greater support for urban climate simulations, especially for small and medium-sized cities worldwide."**

References:

Che, Y., Li, X., Liu, X., Wang, Y., Liao, W., Zheng, X., Zhang, X., Xu, X., Shi, Q., Zhu, J., Zhang, H., Yuan, H., and Dai, Y.: 3D-GloBFP: the first global three-dimensional building footprint dataset, Earth Syst. Sci. Data, 16, 5357-5374, 10.5194/essd-16-5357-2024, 2024.

Cheng, Y., Zhao, L., Chakraborty, T., Oleson, K., Demuzere, M., Liu, X., Che, Y., Liao, W., Zhou, Y., and Li, X.: U-Surf: A Global 1 km spatially continuous urban surface property dataset for kilometer-scale urban-resolving Earth system modeling, Earth Syst. Sci. Data Discuss., 2024, 1-38, 10.5194/essd-2024-416, 2024.

Kamath, H. G., Singh, M., Malviya, N., Martilli, A., He, L., Aliaga, D., He, C., Chen, F., Magruder, L. A., Yang, Z.-L., and Niyogi, D.: GLObal Building heights for Urban Studies (UT-GLOBUS) for city- and street- scale urban simulations: Development and first applications, Sci. Data, 11, 886, 10.1038/s41597-024-03719-w, 2024.

[Figure]

**Figure S5. Pixel-scale comparison of area weighted mean building height between GloUCP and UT-GLOBUS across three representative cities in the United States.** The red dashed line represents the 1:1 line, while the black solid line indicates the fitted regression line.

[Figure]

**Figure S6. Pixel-scale comparison of plan area fraction between GloUCP and UT-GLOBUS across three representative cities in the United States.** The red dashed line represents the 1:1 line, while the black solid line indicates the fitted regression line.

[Figure]

**Figure S7. Pixel-scale comparison of building surface to plan area ratio between GloUCP and UT-GLOBUS across three representative cities in the United States.** The red dashed line represents the 1:1 line, while the black solid line indicates the fitted regression line.

*Does the GloUCP data also include urban fraction parameter that is consistent with the UCP data? If not, I would suggest including the urban fraction data which is very important to be consistent with the derivation of UCP. The authors mentioned they used GAIA data as a mask. Does this mean users can use the urban/impervious fraction data together with GloUCP data? If so, this needs to be clarified.*

**Response**: We sincerely appreciate the reviewer for this constructive comment. Currently, the GloUCP dataset does not include the urban fraction parameter. Based on the reviewer's suggestion, we have added 1 km resolution impervious fraction data to define urban areas in 2020, which is also derived from the GAIA dataset, and shared it alongside the updated GloUCP dataset. This addition not only aligns with our use of the GAIA dataset as a mask but also allows users to conveniently define urban

categories in WRF simulations based on the consistent impervious fraction data. We have added a description of these modifications in Section 2.2 in the revised manuscript.

"**Additionally, to ensure the consistency of the calculation area with the existing impervious surface extent, we further use the Global Artificial Impervious Area (GAIA) dataset in 2020 as a mask for UCP calculation. The GAIA dataset is generated based on long-term optical remote sensing data from the Landsat series of satellites, supplemented by VIIRS nighttime light data and Sentinel-1 radar data (Gong et al., 2020). It uses spatial masking and feature evaluation algorithms to achieve rapid mapping of impervious surfaces, and employs a time consistency verification algorithm to filter and infer logical sequences of impervious surfaces, ensuring their spatial and temporal rationality. In our dataset, only grids with an impervious surface ratio exceeding 1% are retained. Moreover, we have provided 1 km resolution impervious surface fraction data for urban areas in 2020 derived from the GAIA dataset as well. This allows users to conveniently define urban categories (i.e., low-density residential, high-density residential, and industrial/commercial) in WRF simulations based on the consistent impervious fraction data.**"

*Minor comments:*

*Line 65: The reference (Demuzere et al., 2023) for LCZ implementation in WRF is missing. Reference: Demuzere, M., C. He, A. Martilli, and A. Zonato (2023): Technical documentation for the hybrid 100-m global land cover dataset with Local Climate Zones for WRF (1.0.0). Zenodo. https://doi.org/10.5281/zenodo.7670792*

**Response**: We thank the reviewer for this comment. We have incorporated this reference into the manuscript.

*What year is the GloUCP data representative of?*

**Response**: The GloUCP dataset is derived from the 3D-GloBFP dataset in 2020, so it primarily reflects urban morphology for that year. We have mentioned this in the introduction and further explained it in

the data description section in the revised manuscript.

"**However, the latest research has created the first global three-dimensional building footprint dataset (3D-GloBFP) based on publicly available multi-source data (Che et al., 2024). This dataset integrates existing building data to calculate the boundaries and heights of individual buildings globally in 2020.**"

"**Together, these two open-source datasets provide a thorough global spatially continuous building boundary dataset in 2020.**"

*Lines 120-125: It needs to be made clear that these required UCPs are for WRF single layer UCM or all WRF UCMs. WRF includes three types of UCMs (SLUCM, BEP, BEP-BEM). If it is only for single layer UCM, then this should also be made clear in the title.*

**Response**: We thank the reviewer for this comment. Our dataset is designed for three types of Urban Canopy Models (UCMs) in the WRF model: SLUCM, BEP, and BEP-BEM. To provide users with a clearer understanding of the applied UCP schemes, we have elaborated on these parameters in the method and conclusion sections in the revised manuscript, as detailed below:

"**These parameters include mean building height, standard deviation of building height, area weighted mean building height, plan area fraction, building surface to plan area ratio, frontal area index, and distribution of building heights, as detailed in Table 1. They can be applied to three types of UCMs in the WRF model: single-layer urban canopy model (SLUCM), building effect parameterization (BEP), and BEP-BEM (building energy model).**"

"**In this study, we developed a global 1 km spatially continuous UCP dataset — GloUCP, utilizing the latest available building-level information in 2020. It can be applied to all three types of UCMs (i.e., SLUCM, BEP, and BEP-BEM) in the WRF model.**"

**Table 1. Calculation of GloUCP for the WRF model and the applied UCP schemes.**

| Variable | Abbreviation | Formula | Description | Used by UCM (URB_PARAM Index) |
|---|---|---|---|---|
| Mean building height | $\bar{h}$ | $\bar{h} = \frac{1}{N}\sum_{i=1}^{N} h_i$ | $h_i$ is the height of building $i$; $N$ is the total number of buildings in the grid; | SLUCM (92) |
| Standard deviation of building height | $h_{std}$ | $h_{std} = \sqrt{\frac{\sum_{i=1}^{N}(h_i - \bar{h})}{N-1}}$ | | SLUCM (93) |
| Area weighted mean building height | $h_{aw}$ | $h_{aw} = \frac{\sum_{i=1}^{N} A_i h_i}{\sum_{i=1}^{N} A_i}$ | $A_i$ is the plan area on the ground level of building $i$; | SLUCM, BEP, BEP-BEM (94) |
| Plan area fraction | $\lambda_p$ | $\lambda_p = \frac{A_p}{A_T}$ | $A_p$ is the total footprint area of buildings in the grid; $A_T$ is the total area of the grid; | SLUCM, BEP, BEP-BEM (91) |
| Building surface to plan area ratio | $\lambda_b$ | $\lambda_p = \frac{A_R + A_W}{A_T}$ | $A_R$ is the total roof area of buildings in the grid; $A_W$ is the total area of non-horizontal roughness elements (such as walls); | SLUCM, BEP, BEP-BEM (95) |
| Frontal area index | $\lambda_f$ | $\lambda_f(\theta) = \frac{A_{proj}}{A_T}$ | $A_{proj}$ is the total projected area of buildings on a plane perpendicular to four wind directions (0°, 135°, 45°, 90°,); $\theta$ is the wind direction. | SLUCM (96-99) |
| Distribution of building heights | $h_{dis}(i)$ | $h_{dis}(i) = \frac{N_{dis}(i)}{N} \times 100\%$ | $N_{dis}(i)$ is the number of buildings vertically resolved with 5 m bins spanning 0-75 m. | BEP, BEP-BEM (118-132) |

Notes: UCM, urban canopy model; SLUCM, single-layer urban canopy model; BEP, building effect parameterization; BEM, building energy model. The values in parentheses in the last column represent the index of the UCP in the URB_PARAM array.

*Line 125: What year is the GAIA data used in this study representative of? This info is important for regions with fast urbanization rate.*

**Response**: To ensure consistency with the 3D-GloBFP dataset, we used the 2020 impervious surface data from the GAIA dataset. This has been clarified in the revised manuscript.

**"Additionally, to ensure the consistency of the calculation area with the existing impervious surface extent, we further use the Global Artificial Impervious Area (GAIA) dataset in 2020 as a mask for UCP calculation."**

*Does the 3D-GloBFP data also include the Ai, Ap, Ar, Aw, Aproj and Ndis parameters mentioned in Table 1? It is not very clear based on the current descriptions.*

**Response**: The 3D-GloBFP dataset includes only building footprint and height information, and does not provide the Ai, Ap, Ar, Aw, Aproj and Ndis parameters mentioned in Table 1. These parameters were calculated at a 1 km grid level based on building-level information. We have added an explanation of this in the revised manuscript.

**"Overall, the 3D-GloBFP dataset, which can provide global building 2D footprint polygons along with their heights, is the most comprehensive among existing building vector data, making it a robust foundation for calculating UCPs in this study."**

**"In this study, all the UCPs are developed globally at a resolution of approximately 1 km (i.e., 1/120°) based on the building-scale information (i.e., building outline and height) provided by the 3D-GloBFP dataset."**

*I suggest adding a table to summarize the evaluation metrics/statistics for each dataset for Section 3.2.*

**Response**: According to the reviewer's suggestion, we have added two tables in the supplementary materials summarizing the dataset comparisons for three major urban agglomerations in China and

three cities in the United States in the revised manuscript.

**Table S1. Comparison of mean building heights in GloUCP, reference data, and Sun2021 across three major urban agglomerations in China.** Evaluation statistics used are coefficient of determination ($R^2$) and root mean square errors (RMSE).

| Region | Dataset | All | | RMSE (m) | | |
| --- | --- | --- | --- | --- | --- | --- |
| | | $R^2$ | RMSE (m) | height $\leq$ 10 m | 10 < height $\leq$ 24 m | height > 24 m |
| BTH | GloUCP | 0.19 | 14.32 | 7.12 | 4.49 | 33.93 |
| | Sun2021 | 0.09 | 15.22 | 7.39 | 7.08 | 35.38 |
| YRD | GloUCP | 0.20 | 15.88 | 14.02 | 9.02 | 28.93 |
| | Sun2021 | 0.06 | 16.79 | 10.63 | 6.77 | 37.18 |
| GBA | GloUCP | 0.17 | 17.88 | 9.17 | 5.96 | 41.78 |
| | Sun2021 | 0.07 | 19.29 | 12.36 | 8.11 | 42.48 |

**Table S2. Comparison of mean building heights in GloUCP, reference data, and NUDAPT across three representative cities in the United States.** Evaluation statistics used are coefficient of determination ($R^2$) and root mean square errors (RMSE).

| Region | Dataset | All | | RMSE (m) | |
| --- | --- | --- | --- | --- | --- |
| | | $R^2$ | RMSE (m) | height $\leq$ 15 m | height > 15 m |
| Seattle | GloUCP | 0.81 | 2.51 | 1.45 | 8.29 |
| | NUDAPT | 0.64 | 9.03 | 4.15 | 10.72 |
| San Francisco | GloUCP | 0.83 | 4.73 | 2.41 | 7.48 |
| | NUDAPT | 0.73 | 8.57 | 5.78 | 9.22 |
| Philadelphia | GloUCP | 0.52 | 5.50 | 3.58 | 7.14 |
| | NUDAPT | 0.39 | 8.50 | 6.46 | 9.29 |

*Figure 6c,f,i: Should the blue bar be Sun2021 instead of NUDAPT in the legend?*

**Response**: Thank you for pointing out the error. We have corrected Figure 6 accordingly in the revised manuscript.

*One thing that is worth mentioning is that the NUDAPT data in WRF is an old dataset developed around 2010, which contributes to the large uncertainties of this dataset.*

**Response**: Yes, we have also noted the issue of differing data years. Therefore, we have addressed these uncertainties which might introduce into the dataset comparisons in the result section.

"**Nevertheless, using Microsoft's data as reference does not necessarily imply that its height values are absolutely accurate. The heights in Microsoft's dataset were interpolated using a digital terrain model derived from very high-resolution aerial photography, with building boundaries that were hand-digitized. NUDAPT's data were derived from LiDAR measurements, which are also highly accurate. However, it is important to note that the NUDAPT dataset was created using data from the year around 2009, whereas our dataset is based on data from around 2020, leading to a temporal discrepancy. As a result, there is some degree of uncertainty in these comparisons.**"

*I would suggest adding a small section to discuss the potential uncertainties associated with this GloUCP dataset and the cautions/suggestions for users in terms of correctly using this dataset.*

**Response**: According to the reviewer's suggestion, we have added Section 4, "Limitations and uncertainty", to discuss the potential uncertainties associated with the GloUCP dataset in the revised manuscript, as detailed below:

"**4 Limitations and uncertainty**
**The primary purpose of GloUCP is to provide a global 1 km spatially continuous UCPs for three types of UCMs (i.e., SLUCM, BEP, and BEP-BEM) in the WRF model. The uncertainties in**

GloUCP data primarily originate from the 3D-GloBFP dataset, which is generated from multi-source datasets that integrate information with varying spatiotemporal coverage. Additionally, in regions with limited building height samples (e.g., Africa), the accuracy of building height estimation remains relatively low. Therefore, as the accuracy of the 3D-GloBFP dataset improves, the precision of the GloUCP dataset can also be further enhanced.

Considering computational and storage costs, this study only provides UCPs at a global scale with a 1 km resolution. However, based on building vector data, UCP datasets at any spatial resolution can be generated. When regional climate simulations require higher spatial resolution, the resolution of UCP calculations can be adjusted to meet the needs of high-resolution climate modeling. It is important to acknowledge these limitations and uncertainties when using GloUCP data for modeling and analysis. Despite these limitations, GloUCP provides globally comprehensive urban canopy parameters, supporting detailed urban climate simulations on a global scale."

*Is there a plan for the authors to implement this dataset to WRF Pre-processing System (WPS)?*

**Response**: Yes, we have planned to implement our dataset to the WPS after our paper is published.

---

## Author Comment (AC2)

We thank the reviewer for your constructive comments, which significantly improve the quality of our manuscript. Based on these comments, we have revised the manuscript thoroughly. The changes made to the manuscript are noted in the revised manuscript, and described in detail below (reviewer comments are in *italic* and cited texts are in **bold**).

*This article developed the essential data needed to make urban climate simulations based on WRF-Urban as accurate as possible for any city in the world. As the author points out, building morphology data significantly impact the accuracy of urban climate simulations, but such data have only been available for a limited number of cities. This research, which aims to develop a global dataset, is significant in that it attempts to overcome this situation. There is no doubt that the dataset in this paper will be helpful for many WRF-Urban users.*

**Response**: We sincerely thank the reviewer for your positive recognition of our work.

*On the other hand, I am concerned that the paper does not refer to previous research that has undertaken similar initiatives. As a result, from the perspective of an individual data user, it is unclear how these data differ from existing global data and what their characteristics and novelties are. There is a need to compare these data with existing global data and describe the characteristics of this dataset before publication is considered.*

**Response**: We thank the reviewer for your valuable feedback. In response, we have incorporated recent developments in global urban parameter datasets into the revised manuscript, including the global dataset developed by Knanh et al. (2023), the UT-GLOBUS data developed by Kamath et al. (2024), and the U-Surf data developed by Cheng et al. (2024). Based on your constructive comments, we have thoroughly revised the manuscript and hope that our responses and revisions meet your expectations.

References:

Cheng, Y., Zhao, L., Chakraborty, T., Oleson, K., Demuzere, M., Liu, X., Che, Y., Liao, W., Zhou, Y., and Li, X.: U-Surf: A Global 1 km spatially continuous urban surface property dataset for kilometer-scale urban-resolving Earth system modeling, Earth Syst. Sci. Data Discuss., 2024, 1-

38, 10.5194/essd-2024-416, 2024.

Kamath, H. G., Singh, M., Malviya, N., Martilli, A., He, L., Aliaga, D., He, C., Chen, F., Magruder, L. A., Yang, Z.-L., and Niyogi, D.: GLObal Building heights for Urban Studies (UT-GLOBUS) for city- and street- scale urban simulations: Development and first applications, Sci. Data, 11, 886, 10.1038/s41597-024-03719-w, 2024.

Khanh, D. N., Varquez, A. C. G., and Kanda, M.: Impact of urbanization on exposure to extreme warming in megacities, Heliyon, 9, e15511, https://doi.org/10.1016/j.heliyon.2023.e15511, 2023.

*Major comment:*

*As mentioned above, the characteristics of the global dataset proposed in this study must be described in relation to similar existing global datasets.*

*Specifically, we know that a global dataset has been developed and published by Knanh et al. (2023), and these data have been adopted in WRF v4.6.0.*

*--*

*Paper*

*https://www.cell.com/heliyon/fulltext/S2405-8440(23)02718-4*

*Dataset*

*https://figshare.com/articles/dataset/Present_and_future_1_km_resolution_global_population_density_and_urban_morphological_parameters/17108981?file=31635521 (10 Dec 2024, last access)*

*WRF Github*

*https://github.com/wrf-model/WRF/releases/tag/v4.6.0 (10 Dec 2024, last access)*

*https://github.com/wrf-model/WRF/commit/3cadf04277ac3a050e65461efb6aa939349c60a8 (10 Dec 2024, last access)*

*https://github.com/wrf-model/WRF/pull/1986 (10 Dec 2024, last access)*

*--*

**Response**: We sincerely thank the reviewer for the constructive suggestions. Regarding the recently released datasets mentioned above, the U-Surf dataset developed by Cheng et al. (2024) and our GloUCP dataset are both derived from the same building vector data released by Che et al. (2024) to calculate urban morphological parameters. As a result, there is no difference in mean building height

between the two datasets. We have clarified this in the revised manuscript and further explained that the urban morphological parameters provided by U-Surf, which are designed for the UCM in the Community Earth System Model, differ from the UCPs required by WRF/UCM and therefore cannot be directly applied in the WRF model. We highlight that the uniqueness of our study lies in utilizing a newly-developed building-scale height map to produce a global, spatially continuous, high-resolution UCP dataset specifically tailored for the WRF model.

Additionally, based on the reviewer's suggestion, we compared our dataset with the recently released UT-GLOBUS dataset by Kamath et al. (2024) and the global urban morphological dataset developed by Khanh et al. (2023). For the UT-GLOBUS dataset, our dataset offers more comprehensive spatial coverage, as the UT-GLOBUS dataset has significant data gaps, particularly in regions such as East Asia. We also conducted a comparison of our GloUCP dataset with the UT-GLOBUS dataset across three cities in the United States. Overall, compared to the reference data, our dataset and the UT-GLOBUS dataset demonstrate similar levels of accuracy (Figs. S5–S7).

For the global urban morphological dataset developed by Khanh et al. (2023), we used their UCPs estimated based on GDP and population density information in 2010 (hereafter referred to as Knanh2010) to compare it with our data in representative regions in China and the United States, respectively. The results show that our dataset still has an advantage in terms of spatial coverage (Figs. S8 and S9). In terms of accuracy, the Knanh2010 dataset performs well in capturing low- to mid-rise buildings but significantly underestimates the height of high-rise buildings in China (Fig. S10). In contrast, both datasets perform reasonably well in U.S. cities, although our dataset shows slightly better accuracy across different building height categories (Fig. S11).

In summary, we have added a brief description of the above-mentioned datasets in the introduction section and highlighted the advantages of GloUCP compared to these recently released UCP datasets. Furthermore, we included a detailed comparison between our dataset and existing global datasets in the results section to emphasize their differences in the revised manuscript, as detailed below.

[revised manuscript text omitted]

*Minor comment:*

*1. I think the authors developed data for both the single-layer urban canopy model (SLUCM) and the Multi-Layer UCM (BEP). If my understanding is correct, it would be better to describe this kind of information; otherwise, readers might misunderstand. If I understand correctly, a global dataset of the vertical distribution of building heights (for BEP and BEP+BEM) would be a novelty of this work.*

**Response**: We thank the reviewer for this comment. Our dataset is indeed developed for both the single-layer urban canopy model (SLUCM) and the multi-layer UCM. In Table 1, we categorized and explained the parameters based on the requirements of the three types of UCMs in the WRF model: the single-layer urban canopy model (SLUCM), the Building Effect Parameterization (BEP), and the BEP-BEM (Building Energy Model). Additionally, we further emphasized in method and conclusion sections in the revised manuscript that our dataset provides parameters suitable for both the SLUCM and the multi-layer UCM.

"**These parameters include mean building height, standard deviation of building height, area weighted mean building height, plan area fraction, building surface to plan area ratio, frontal area index, and distribution of building heights, as detailed in Table 1. They can be applied to three types of UCMs in the WRF model: single-layer urban canopy model (SLUCM), building effect parameterization (BEP), and BEP-BEM (building energy model).**"

"**The primary purpose of GloUCP is to provide a global 1 km spatially continuous UCPs for three types of UCMs (i.e., SLUCM, BEP, and BEP-BEM) in the WRF model.**"

"**In this study, we developed a global 1 km spatially continuous UCP dataset — GloUCP, utilizing the latest available building-level information in 2020. It can be applied to all three types of UCMs (i.e., SLUCM, BEP, and BEP-BEM) in the WRF model.**"

**Table 1. Calculation of GloUCP for the WRF model and the applied UCP schemes.**

| Variable | Abbreviation | Formula | Description | Used by UCM (URB_PARAM Index) |
|---|---|---|---|---|
| Mean building height | $\bar{h}$ | $\bar{h} = \frac{1}{N}\sum_{i=1}^{N} h_i$ | $h_i$ is the height of building $i$; $N$ is the total number of buildings in the grid; | SLUCM (92) |
| Standard deviation of building height | $h_{std}$ | $h_{std} = \sqrt{\frac{\sum_{i=1}^{N}(h_i - \bar{h})}{N-1}}$ | | SLUCM (93) |
| Area weighted mean building height | $h_{aw}$ | $h_{aw} = \frac{\sum_{i=1}^{N} A_i h_i}{\sum_{i=1}^{N} A_i}$ | $A_i$ is the plan area on the ground level of building $i$; | SLUCM, BEP, BEP-BEM (94) |
| Plan area fraction | $\lambda_p$ | $\lambda_p = \frac{A_p}{A_T}$ | $A_p$ is the total footprint area of buildings in the grid; $A_T$ is the total area of the grid; | SLUCM, BEP, BEP-BEM (91) |
| Building surface to plan area ratio | $\lambda_b$ | $\lambda_p = \frac{A_R + A_W}{A_T}$ | $A_R$ is the total roof area of buildings in the grid; $A_W$ is the total area of non-horizontal roughness elements (such as walls); | SLUCM, BEP, BEP-BEM (95) |
| Frontal area index | $\lambda_f$ | $\lambda_f(\theta) = \frac{A_{proj}}{A_T}$ | $A_{proj}$ is the total projected area of buildings on a plane perpendicular to four wind directions (0°, 135°, 45°, 90°,); $\theta$ is the wind direction. | SLUCM (96-99) |
| Distribution of building heights | $h_{dis}(i)$ | $h_{dis}(i) = \frac{N_{dis}(i)}{N} \times 100\%$ | $N_{dis}(i)$ is the number of buildings vertically resolved with 5 m bins spanning 0-75 m. | BEP, BEP-BEM (118-132) |

Notes: UCM, urban canopy model; SLUCM, single-layer urban canopy model; BEP, building effect parameterization; BEM, building energy model. The values in parentheses in the last column represent the index of the UCP in the URB_PARAM array.

*2. It would be helpful for users if you added a table showing the relationship between the parameters in the dataset you developed and the parameters in URBPRAM.TBL in WRF-Urban.*

**Response**: Following the reviewer's suggestion, we have added relevant content to Table 1 in the revised manuscript to explain the correspondence between the parameters we calculated and those in URBPARM.TBL.

*3. I wonder about the correspondence between the parameters of the dataset developed this time and the categories of the global map of Global LCZ. Global LCZ is included in WRF and it is necessary to set geometric parameters etc. for each LCZ category in URBPRAM.TBL. I am therefore concerned about the above correspondence. Is the proposed dataset intended to set values for each WRF grid? Or is it intended to set values according to LCZ? It would be helpful to users if you could describe the author's thinking and recommended setting methods.*

**Response**: We thank the reviewer for this comment. Our dataset is designed to set UCPs for each WRF grid. Currently, in the urban canopy model, urban can be classified either into three categories (i.e., low-density residential, high-density residential, and industrial/commercial), or using LCZ classifications. However, each urban type can only be assigned a single fixed set of UCP values. Actually, when the urban type of each grid is determined, our dataset can be used to reassign urban morphological parameters for each grid, thereby providing a more detailed and accurate depiction of urban morphological variations within the study areas. We have added a description of these modifications in Section 2.2 in the revised manuscript

"**Moreover, we have provided 1 km resolution impervious surface fraction data for urban areas in 2020 derived from the GAIA dataset as well. This allows users to conveniently define urban categories (i.e., low-density residential, high-density residential, and industrial/commercial) in WRF simulations based on the consistent impervious fraction data. Once the urban type of each grid is determined, our dataset can be used to reassign urban morphological parameters for each grid, thereby providing a more detailed and accurate depiction of urban morphological variations within the study areas.**"

---

## Author Response (AR2)

**Referee #2**

I would like to thank the author for the detailed and careful revisions. I now better understand the differences in positioning and characteristics compared to existing datasets. I think this is a helpful revision for readers as well.

**Response**: We sincerely appreciate the reviewer's positive feedback and recognition of our revisions. Thank you again for your valuable suggestions, which have helped improve the clarity and completeness of our work. The changes made to the manuscript are noted in the revised manuscript, and described in detail below (reviewer comments are in *italic* and cited texts are in **bold**).

**There is one additional question I would like to ask.**

In the comparison with Knanh2010, I could understand that in Chinese cities, Knanh2010 had less reproducibility for high-rise buildings, while in US cities this feature was not seen. Is it possible to consider possible reasons for this, whether because the data sources used in Knanh2010 and this study are different (including the year of the source), or because the estimation methods are different? For example, because Knanh2010 uses data sources from 2010, it does not capture the rapid development of Chinese cities (construction of high-rise buildings) after 2010, etc. I don't think it's necessary to explore this in detail in this study, but I think it would be helpful for readers if you mentioned it a little in your discussion.

**Response**: We thank the reviewer for the constructive suggestions. As you pointed out, these differences may be due to variations in data sources (including the reference year) and estimation methods. Knanh2010 is based on 2010 data, which may not fully capture the rapid high-rise development in China after that period, whereas our dataset is derived from more recent building-scale data. Additionally, Knanh2010 uses GDP and population density-based estimations, while our dataset relies on direct building footprint and height data, which could further explain the discrepancy. We have briefly mentioned this point in the discussion section, as suggested.

"In China, compared to the reference data, the Knanh2010 dataset performs well in capturing low- to mid-rise buildings but significantly underestimates the height of high-rise buildings (Fig. S10). However, this issue is not as prominent in U.S. cities, where the accuracy of our GloUCP dataset and Knanh2010 is relatively similar, with our dataset performing slightly better across different building height categories (Fig. S11). This discrepancy in reproducibility between Chinese and U.S. cities may be attributed to differences in data sources (including the reference year) and estimation methods used in Knanh2010 and this study. Since Knanh2010 is based on data from 2010, it may not fully capture the rapid urban expansion and the widespread construction of high-rise buildings that occurred in China after 2010. In contrast, our dataset, derived from more recent data sources, better reflects contemporary urban morphology. Additionally, differences in the estimation approaches-Knanh2010 relying on GDP and population densitybased empirical models, while our dataset is constructed using building-scale vector data-could also contribute to these variations. Overall, while the Knanh2010 dataset already offers better spatial coverage than most existing datasets, our GloUCP dataset provides even more comprehensive coverage. In China, the accuracy of most datasets remains suboptimal, but our dataset slightly outperforms others, particularly in representing high-rise buildings."